# Assessing the Impact of Influenza Vaccination Timing on Experimental Arthritis: Effects on Disease Progression and Inflammatory Biomarkers

**DOI:** 10.3390/ijms25063292

**Published:** 2024-03-14

**Authors:** Vera Tarjányi, Ákos Ménes, Leila Hamid, Andrea Kurucz, Dániel Priksz, Balázs Varga, Rudolf Gesztelyi, Rita Kiss, Ádám István Horváth, Nikolett Szentes, Zsuzsanna Helyes, Zoltán Szilvássy, Mariann Bombicz

**Affiliations:** 1Department of Pharmacology and Pharmacotherapy, Faculty of Medicine, University of Debrecen, H-4032 Debrecen, Hungary; tarjanyi.vera@med.unideb.hu (V.T.); akos.menes@gmail.com (Á.M.); leilahamid@hotmail.com (L.H.); varga.balazs@pharm.unideb.hu (B.V.); gesztelyi.rudolf@pharm.unideb.hu (R.G.); kiss.rita@med.unideb.hu (R.K.); szilvassy.zoltan@med.unideb.hu (Z.S.); 2Cardiology and Cardiac Surgery Clinic, University of Debrecen, H-4032 Debrecen, Hungary; kurucz.andrea@med.unideb.hu; 3Department of Pharmacology and Pharmacotherapy, Medical School, University of Pécs, H-7624 Pécs, Hungary; adam.horvath@aok.pte.hu (Á.I.H.); szentes.nikolett@gmail.com (N.S.); zsuzsanna.helyes@aok.pte.hu (Z.H.); 4National Laboratory for Drug Research and Development, H-1117 Budapest, Hungary; 5Hungarian Research Network (HUN-REN-PTE), Chronic Pain Research Group, Medical School, University of Pécs, H-7624 Pécs, Hungary

**Keywords:** vaccination, influenza, complete Freund’s adjuvant-induced, autoimmunity, inflammation, reactive oxygen species, neutrophil myeloperoxidase activity, heme oxygenase-1, vaccine-mediated autoimmunity, adjuvant effect

## Abstract

Numerous studies have indicated a link between vaccines and the exacerbation of autoimmune diseases including rheumatoid arthritis (RA). However, there is no consensus in clinical practice regarding the optimal timing of immunization. Therefore, this study aimed to investigate the impact of the 3Fluart influenza vaccine on the complete Freund’s adjuvant (CFA)-induced chronic arthritis rat model and to identify new biomarkers with clinical utility. CFA was injected into the plantar surface of one hind paw and the root of the tail on day 0, and the tail root injection was repeated on day 1. Flu vaccination was performed on day 1 or 7. Paw volume was measured by plethysmometry, mechanonociceptive threshold by dynamic plantar aesthesiometry, neutrophil myeloperoxidase (MPO) activity, and vascular leakage using in vivo optical imaging throughout the 21-day experiment. Inflammatory markers were determined by Western blot and histopathology. CFA-induced swelling, an increase in MPO activity, plasma extravasation in the tibiotarsal joint. Mechanical hyperalgesia of the hind paw was observed 3 days after the injection, which gradually decreased. Co-administration of the flu vaccine on day 7 but not on day 1 resulted in significantly increased heme oxygenase 1 (HO-1) expression. The influenza vaccination appears to have a limited impact on the progression and severity of the inflammatory response and associated pain. Nevertheless, delayed vaccination could alter the disease activity, as indicated by the findings from assessments of edema and inflammatory biomarkers. HO-1 may serve as a potential marker for the severity of inflammation, particularly in the case of delayed vaccination. However, further investigation is needed to fully understand the regulation and role of HO-1, a task that falls outside the scope of the current study.

## 1. Introduction

The COVID-19 pandemic brought the debate about vaccines back to the forefront of both scientific and political discussions. Vaccines remain one of the most effective tools for the primary prevention of several infectious diseases. Therefore, it is essential for the scientific community to maintain trust in vaccines, which has been shaken in some cases [1]. Several mechanisms of autoimmunity and immune tolerance still remain to be explored. Since infections are potential triggers for autoimmune reactions, similar effects from vaccines cannot be excluded [2]. They usually contain some microbial particles, which effectively activate immune cells leading to increased intercellular communication. These complex interactions can lead to unwanted consequences due to molecular mimicry and bystander activation [3]. A recent review suggests that autoimmune side effects are not remarkable in cases of the most currently used vaccines, but further research is encouraged [4,5].

Rheumatoid arthritis (RA) is a prevalent systemic autoimmune disease with synovial inflammation that can result in severe and irreversible joint damage, causing significant disability [6,7]. The inflammatory situation in the synovium maintains abundant reactive oxygen species (ROS) production which can lead to oxidative stress. High amounts of ROS produced by phagocytes, recruited immune cells, and proliferating synovial stromal cells could play a key role in the pathogenesis of RA. ROS-scavenging antioxidants can lead to the early and more efficient treatment of RA patients [8].

The adjuvant-induced arthritis (AIA) model is commonly used with rodents to mimic the main pathologic features and the progression of RA. Complete Freund’s adjuvant (CFA) contains heat-killed Mycobacterium tuberculosis suspended in paraffin oil. The cell wall of the bacteria contains muramyl dipeptide which can cause macrophage-driven T-helper1 (Th1, CD4+) lymphocyte activation leading to chronic destructive arthritis [9]. AIA develops in two phases as follows: an acute articular inflammation followed by a late phase with bone involvement [10]. At the site of injection, a reddish and swollen continuous inflammation is observed, which begins as early as 2 h after the injection and peaks within 8 h. This acute inflammatory phase lasts for 3–4 days. Four days after CFA administration, the levels of the erythrocyte sedimentation rate (ESR), blood neutrophils, and leukocyte counts start to increase. Due to the extensive infiltration of neutrophils and the proliferation of the synovial lining, hyperalgesia and edema develop in the ankle and dorsal region of the tarsus at weeks 1–2. This inflammation starts to affect the adjacent joints as macrophages release pro-inflammatory cytokines such as tumor necrosis factor-α (TNF-α), interferon γ (IFNγ), interleukin (IL)-6, and IL-12-activating CD4+ T cells [11]. According to the disease progression, we have chosen two time points for the flu vaccination as follows: one at the beginning of the acute phase (on day 1) and one at the beginning of the chronic phase (on day 7).

While our understanding of cytokines continues to advance, there remains conflicting data among researchers regarding their validity as biomarkers for disease. Diagnostic biomarkers, though widely utilized and endorsed by the ACR/EULAR recommendations, are not all specific indicators of inflammation (Table 1). Such numbers indicate a noticeable lack of clinical studies focusing on exploring the cross-talk between oxidative stress and RA. Consequently, current research highlights oxidative stress as a significant area of study for identifying the biomarkers for RA [12]. Heme oxygenase-1 (HO-1), a stress protein and metabolic enzyme, remains a subject of global interest in both basic and translational research due to its role in regulating cellular and tissue homeostasis, modulating immune function, and reducing inflammation. The generation of heme-derived reaction products (such as biliverdin and bilirubin) could potentially enhance HO-dependent cytoprotection through the antioxidant and immunomodulatory effect [13]. Furthermore, there is a clear association between the induction of heme oxygenase-1 and a reduction in the expression of the numerous inflammatory cytokines observed for rheumatoid arthritis [14].

Influenza is an acute viral respiratory disease that potentially leads to hospitalizations and even deaths during annual winter epidemics. Yearly vaccinations are required against influenza A and influenza B, which are inactivated preparations containing subunit or subvirion (split) surface antigens. It has been described that both influenza virus infection and influenza vaccination may be implicated in autoimmune complications [36].

Based on the limited knowledge on the safety and immunogenicity of vaccinations, the major concern of the updated European Alliance of Associations for Rheumatology (EULAR) statement is that influenza vaccination should be strongly considered for the majority of patients with autoimmune inflammatory rheumatic diseases (AIIRDs) since they have a higher risk of influenza compared to that of the healthy population [37,38]. A prospective vaccination study from 2011 suggests that the time of vaccination has a high impact on disease progression [39]. A retrospective, nationwide study showed that vaccination lowers morbidity and mortality for RA patients, especially for the older population [40]. Generally, in line with these findings, immunization should be preferentially administered during the quiescent phase of disease. However, in an active disease, immunization should not be precluded, but individual-based decision should be considered [37].

Altogether, limited data have been published on vaccine-associated RA exacerbation. Therefore, in this study, we examined the 3Fluart influenza vaccine in a rat model of RA to explore the links between vaccination and disease progression. We aim to provide a safe protocol for vaccination in harmonization with disease activity as well as to identify potential new diagnostic/prognostic biomarkers.

## 2. Results

### 2.1. Influenza Vaccination Has No Effect on Body Weight Changes

There were no observable differences between the CFA-injected groups; they all had a lower body weight gain compared to that of the respective non-arthritic controls.

On the 21st day of the experiment, the body weight changes in the Sham Flu early group were higher than in the CFA Flu early group (Figure 1A,B, Appendix A).

### 2.2. Influenza Vaccination Has No Effect on CFA-Induced Paw Edema and Mechanical Allodynia

CFA induced a significant paw volume increase in all the CFA-injected groups compared to those observed for the respective Sham groups from the beginning of the study (on day 1, the difference between the Sham Control and CFA Control was 69%), which was maintained until day 14. However, flu vaccination had no effect on CFA-induced paw edema at any of the timepoints (Figure 2A, Appendix A).

CFA induced a significant mechanonociceptive threshold decrease (mechanical allodynia) in all CFA-injected groups compared to that of the the Sham groups from the beginning of the study (on day 1, the mechanonciceptive threshold of the CFA Control group was 63% lower than that measured in the respective Sham group), which was maintained until the end of the experiment (day 21). However, flu vaccination had no effect on CFA-induced mechanical allodynia at any of the timepoints (Figure 2B, Appendix A).

### 2.3. Influenza Vaccination Does Not Influence the Neutrophil Myeloperoxidase (MPO) Activity in the CFA-Injected Hind Paws

Luminol-derived bioluminescence showed a significant increase in neutrophil MPO activity in the ipsilateral hind paws of all CFA-injected rats 2 days after arthritis induction compared to that of the respective non-arthritic controls. The luminescent signal further increased in the arthritic groups until day 9, but a significant difference was not observed compared to that of day 2. Fluart vaccines did not influence an increase in CFA-induced neutrophil MPO activity at any of the time points (Figure 3A,B and Appendix A).

### 2.4. Influenza Vaccination Does Not Influence Plasma Extravasation in the CFA-Injected Tibiotarsal Joints

IR-676-derived fluorescence showed a significant plasma extravasation increase from the leaky venules in the ipsilateral hind paws of all the CFA-injected rats 2 days after arthritis induction compared to that of the respective non-arthritic controls. The fluorescent signal increased further in the arthritic groups until day 9, which was significant only in the case of the non-vaccinated CFA-injected group compared to day 2. Between the vaccinated and the non-vaccinated arthritic groups, a significant difference was not observed at any of the time points (Figure 4A,B and Appendix A).

### 2.5. Influenza Vaccination Has No Effect on the CFA-Induced Inflammatory Biomarker Expression Increase

Regarding the MPO expression, no significant difference was observable due to the effects of CFA administration alone (Sham Control vs. CFA Control); however, the relative expression of neutrophil MPO was the highest in the CFA Flu late group, and this difference was significant compared to that of the Sham Flu early, Sham Flu late, and Sham Control groups. Examining the expression of the HO-1 enzyme, we observed that the CFA injection alone likewise had no effect on its expression. It was the highest in the CFA Flu late group, and the difference was significant compared to that of the Sham Flu late group. Related to the matrix metalloproteinase 9 (MMP9) expression, CFA administration alone elevates it but not in a significant manner (Sham Control vs. CFA Control). Nevertheless, the CFA Flu late group presented a higher MMP9 expression compared to the Sham Control, Sham Flu early, and Sham Flu late groups. No significant differences were detected relating to the relative expression of TNF-α (Figure 5, Appendix A).

## 3. Discussion

To our knowledge, this is the first study to investigate the impact of the flu vaccine using a rat arthritis model, revealing that administering the vaccine causes changes in some potential biomarkers for assessing inflammation severity in experimental arthritis.

Vaccines are preventive treatment options for infections with high morbidity and mortality. Vaccinations are sometimes associated with hypersensitivity and autoimmunity that can be severe and fatal [41]. It has been shown that infective agents can provoke autoimmune diseases in a prone subject through various mechanisms including, but not limited to, epitope spreading, polyclonal activation, and molecular mimicry. Some claim that the most widely used aluminum hydroxide and phosphate adjuvant components of vaccines trigger autoimmunity, which is known as autoimmune/inflammatory syndrome induced by adjuvants (ASIA) [42]. Our hypothesis was that the timing of vaccination may highly affect the course of an autoimmune disease, and immunization during the early stages determined by a highly sensitive biomarker could substantially mitigate the risk of adverse effects and enhance the safety of flu vaccination for RA patients.

Consistent with previous studies, CFA significantly reduced body weight gain and induced mechanical allodynia as well as paw edema in rats. [43,44]. Although body weight alone does not qualify as a selective marker for determining CFA-induced arthritis, it is a relevant functional parameter which is comparable to previous studies. Meanwhile, hyperalgesia and allodynia in arthritis are at least partially related to peripheral inflammation, where several cytokines, prostaglandins, and proteolytic enzymes are responsible for complex mechanisms [45]. The maximum of the CFA-induced tibiotarsal joint swelling was observed at around the 11th day of the experiment [27], which was not altered by the flu vaccinations.

Non-invasive imaging methods specific to different inflammatory mechanisms allow for longitudinal quantitative assessment of the disease. We have previously demonstrated the usefulness of the chemiluminescent agent luminol and a near-infrared cyanin dye (IR-676) for non-invasive imaging of the cellular and vascular components, respectively. Luminol enables visualization of neutrophil MPO activity, and IR-676 can be used to assess vascular hyperpermeability [46]. In our experiment, plasma protein extravasation increased on day 2 after CFA injection, which further intensified on day 9. However, flu vaccination did not alter plasma extravasation at any time point.

A significant elevation in MPO activity was observed in the tibiotarsal joints of both the CFA-injected groups on day 2, which was maintained at the beginning of the chronic phase on day 9. However, flu vaccination did not alter in vivo MPO activity at any time point. In agreement with the in vivo data, Western blot results of MPO expression in the excised joints increased after the CFA injection on day 21. However, MPO protein expression was still slightly elevated in the arthritic joints of the early-vaccinated rats but was markedly elevated in the late-vaccinated rats. As seen from the edema formation, vaccination in the later stage also resulted in a more pronounced inflammatory response triggered by the adjuvant compared to that of the Sham Control group.

During the earlier phase of inflammation, leukocytes are the most abundant cell types in RA, and they fuel inflammation by releasing multiple proteolytic enzymes, including MMPs. Neutrophils produce neutrophil collagenase (MMP-8) and gelatinase B (MMP-9), which metabolize various types of collagens (IV, V, VII, VIII, IX), elastin, and fibronectins in the joint, contributing to histological alterations such as synovial enlargement, cartilage erosion, and bone destruction [47,48,49,50]. According to our Western blot results on day 21, MMP-9 showed a similar pattern to that of MPO. Specifically, a marked expression was observed in the joints of the CFA-treated rats compared to those of the Sham Controls, and the same elevation was seen in both flu-vaccinated arthritic groups. However, in the late flu-vaccinated group, the protein expression highly increased compared to that of the CFA Controls. Firstly, these results support the notion that MPO and MMP-9 levels both indicate increased neutrophil and macrophage activity. Secondly, this inflammation is more pronounced with later flu immunization. Alterations in the joint extracellular matrix turnover is an important factor of the local inflammatory symptoms in RA [51]. The degradation of joint cartilage, regulated by matrix metalloproteinases, is under the control of cytokines, primarily TNF-α and interleukin-1, which enhance the synthesis of MMP-9 [52,53]. We see a significant elevation in the HO-1 protein expression in the joint of the late-vaccinated arthritic group. HO-1 is an inducible form of HO, also called heat shock protein-32. It is upregulated after internal or exogenous stimuli such as bacterial lipopolysaccharides, oxidative stress, ischemia, reperfusion, or being in the presence of hemin, which is the enzyme’s main substrate [54]. HO-1 has numerous cytoprotective effects by degrading heme into biliverdin, ferrous ion (Fe^2+^), and carbon monoxide (CO). CO is known to suppress the synthesis of inflammatory mediators (cytokines, nitric-oxide, prostaglandins), and bilirubin (after reduction from biliverdin) as well as ferritin can reduce the signs of inflammation by being antioxidants [55]. Overall, the upregulation of HO-1 in response to oxidative stress and inflammation in RA can modulate neutrophil activity and function. HO-1 can be a marker of the enhanced inflammatory process in the joints.

In summary, CFA-induced chronic arthritis and allodynia are not markedly influenced by either early or late flu vaccination; therefore, it is suggested to be relatively safe for experimental arthritis. However, joint inflammation through edema formation and the release of multiple inflammatory and pro-oxidant markers, such as MPO and MMP-9, is at least partially increased by late flu vaccination. If oxidative stress and inflammation increase, the HO-1 defense pathway may be activated. Therefore, HO-1 could be considered as a biomarker indicating the severity of inflammation in RA.

## 4. Materials and Methods

### 4.1. Animals

Our experiment was performed on 33 male Lewis rats (Charles River, Budapest, Hungary). The animals were kept in the Laboratory Animal House of the University of Pécs in 375 × 215 × 180 mm sized cages, with a maximum of 2 rats per cage under a 12-h light/dark cycle at 24 ± 2 °C with 50–60% humidity. The animals were supplied with water ad libitum, and they were fed with ordinary rat chow (ssniff Spezialdiäten, Soest, Germany). Before all of the applied methods, an acclimatization period was applied, which allowed the animals to become used to the present conditions.

### 4.2. Induction of the Experimental Disease

Chronic arthritis was induced by injecting CFA (heat-killed Mycobacterium tuberculosis suspended in paraffin oil, 1 mg/mL, Sigma Aldrich, St. Louis, MO, USA) subcutaneously into the plantar surface of the hind paw (50 μL) and the root of the tail (50 μL) (day 0). To enhance the systemic effects, we repeated the injection into the tail root (50 μL) on the next day, which was considered to be the first day of the experiment [50].

### 4.3. Treatment

The animals were divided into six groups depending on the treatment administered. As a Sham group, we considered six rats, which did not receive any therapy or CFA injection. The CFA Flu early group (*n* = 7) received a single intramuscular dose (0.5 mL) of a commercially available vaccine (The Fluart Innovative Vaccines Ltd., Pilisborosjenő, Hungary) on the first day of the experiment, in contrast with the CFA Flu late group members (*n* = 7), which were administered the vaccine on the seventh day. Vaccines contained 6 µg of inactivated purified surface fragments from each of the three different strains of the influenza virus [A/Michigan/45/2015 (H1N1) pdm09-like virus, A/Singapore/INFIMH-16-0019/2016 A(H3N2)-like virus (updated), B/Colorado/06/2017-like (Victoria lineage) virus (updated)] according to the recommendations of the World Health Organization (WHO) for the winter season of 2018–2019. Other vaccine ingredients were aluminum chloride hexahydrate, trisodium phosphate dodecahydrate, potassium chloride, thiomersal, disodium hydrogen phosphate dihydrate, potassium dihydrogen phosphate, sodium chloride, and water for injections. The vaccine contained an adjuvant, which was aluminum phosphate gel (max. 0.625 milligrams Al^3+^). We created a CFA Control group too; they were injected only with CFA (*n* = 7). There were two groups injected only with the vaccine either on the first or the seventh day of the experiment (Sham Flu early (*n* = 7), Sham Flu late (*n* = 6)) (Table 2). The study was terminated on the 21st day by over anesthesia of the animals with sodium-pentobarbital (Euthanimal, Alfasan Nederland B.V., Woerden, The Netherlands, 100 mg/kg, i.p.).

### 4.4. Measurement of Body Weight

To evaluate changes in body weight caused by the CFA injection and vaccination, animals were weighed at the beginning of the experiment and then on the 2nd, 9th, and 21st days.

### 4.5. Measurement of Paw Edema

For measuring the changes in the paw volume, plethysmometry was used (Ugo Basile, Gemonio, Italy) at the beginning of the experiment and on the 1st, 4th, 11th, and 18th days of the experiment. This method works based on the principles of communicating vessels. The device has two vessels filled with fluid. During the procedure, one paw of the animal is dipped into one of the vessels until a previously determined level while the transducer placed in the other vessel measures the volume of the fluid squeezed out, which is displayed in cm^3^. The changes in the paw volume are given as a percentage compared with the control values [56].

### 4.6. Measurement of Mechanonociceptive Threshold

Mechanonociception of the animals was evaluated using dynamic plantar aesthesiometry (Ugo Basile, Gemonio, Italy). The animals were placed into a 15 × 15 cm sized cage with a metal mesh surface. During the procedure, a straight metal filament touched the plantar surface of the rats with an increasing force (5 g/s) until it reached 50 g or until the rat showed a withdrawal reaction. When the rat removed its paw, the machine stopped immediately, thus avoiding further inconvenience or tissue damage, and the numerical value of the allodynia was readable from the digital display. There were three measurements on both paws, and further investigation was conducted using the mean of these values. The decrease in the mechanonociceptive threshold (mechanical allodynia) was given as a percentage relative to the control values measured before the first day of the experiment [57]. The measurements were conducted at the beginning of the experiment and then on the 1st, 4th, 7th, 14th, and 21st days.

### 4.7. In Vivo Bioluminescence Imaging of Neutrophil MPO Activity

The activity of the neutrophil MPO, which is an important factor in the pathomechanism of RA, was evaluated by in vivo luminescence imaging using the IVIS Lumina III In Vivo Imaging System (Perkin Elmer, Waltham, MA, USA) [58]. As a contrast agent, luminol sodium salt (5-Amino-2,3-dihydrophthalazine-1,4-dione, Gold Biotechnology, Olivette, MO, USA), which is a chemiluminescence compound, was applied specifically for the neutrophil MPO activity and for the ROS produced by MPO. During the procedure, the animals were anesthetized with ketamine/xylazine (Calypsol, Richter Gedeon Nyrt., Budapest, Hungary; Sedaxylan, Eurovet Animal Health B.V., Bladel, The Netherlands, 100/10 mg/kg, ip.) and ip. injected with 150 mg/kg luminol dissolved in sterile phosphate-buffered saline (PBS, 30 mg/mL). The value of bioluminescence was measured 10 min postinjection. It was given as the total radiance (total photon flux/s) on assigned areas above the tibiotarsal joints with Living Image^®^ software 4.5.2. (Perkin Elmer, Waltham, MA, USA).

### 4.8. In Vivo Fluorescence Imaging of Vascular Leakage

The value of vascular leakage was evaluated by fluorescence in vivo imaging using the IVIS Lumina III In Vivo Imaging System (PerkinElmer, Waltham, MA, USA). IR-676 (Spectrum-Info Ltd., Kyiv, Ukraine) was used as the contrast agent, which is suitable for imaging inflammatory hypervascularization and vascular leakage [59]. Prior to examination, the animals were anesthetized with ketamine/xylazine (Calypsol, Richter Gedeon Nyrt., Budapest, Hungary; Sedaxylan, Eurovet Animal Health B.V., Bladel, the Netherlands, 100/10 mg/kg, ip.), and a 0.5 mg/kg dose of fluorophore was administered intravenously (i.v.) in a 5 *v*/*v*% Kolliphor HS 15 (Sigma Aldrich, St. Louis, MO, USA) solution, which acts as a stabilizer for IR-676. Fluorescence measurements were taken 20 min postinjection. The fluorescence value was expressed as the total radiant efficiency ([photons/s/cm^2^/sr]/[µW/cm^2^]) and measured above the tibiotarsal joints in a pre-defined area of interest (Regions of Interest, ROIs) using Living Image^®^ software 4.5.2. (Perkin Elmer, Waltham, MA, USA).

### 4.9. Western Blot

For the analysis of the tibiotarsal joints, Western blot analysis was performed. Deep-frozen samples (300 mg at −80 °C) were treated with liquid nitrogen and homogenized using a Polytron-homogenizer (IKA-WERKE, Staufen, Germany) in 800 µL buffer (25 mM Tris-HCl, pH = 8, 25 mM NaCl, 1 mM Na-orthovanadate, 10 mM NaF, 10 mM Na-pyrophosphate, 10 nM okadaic acid, 0.5 mM EDTA, 1 mM PMSF, and protease inhibitor cocktail (Sigma-Aldrich, St. Louis, MO, USA)). After centrifugation (at 2000 rpm for 10 min and 4000 rpm for 2 min), 250 μL of Triton X-containing solvent was added to each homogenized sample, followed by a one-hour incubation period on ice. The homogenization procedure was completed with another round of centrifugation (14,000 rpm for 10 min). The supernatant containing both the cytosolic and mitochondrial compartments, was used to determine the total protein concentration. The QuantiPro™ BCA Assay Kit (Sigma-Aldrich-Merck KGaA, Darmstadt, Germany) was used to select samples with the appropriate amount of total protein (20 μg). Gel electrophoresis (using 12% gel) was performed at 40 mA for 100–120 min to separate proteins based on their molecular weight. Protein transfer onto a nitrocellulose membrane (GE Healthcare, New York, NY, USA) was achieved by electro-blotting at 25 V for 90 min, followed by 1 h of blocking at room temperature in 3% BSA-containing TBS-T.

The membranes were hen-incubated overnight with the following antibodies purchased from Sigma-Aldrich (Sigma-Aldrich-Merck KGaA) and Abcam (Abcam Plc., Cambridge, UK): myeloperoxidase (MPO), heme oxygenase-1 (HMOX1), matrix-metalloproteinase 9 (MMP9), and tumor necrosis factor-alpha (TNF-α). The antibodies were applied according to the manufacturer’s recommendation. After the first incubation with primary antibodies, a second incubation with secondary antibodies was performed. For protein detection, labeling was carried out with horseradish peroxidase, and the membranes were scanned using a C-Digit© blot scanner with Image Studio Digits ver. 5.2. software (LI-COR Inc., Lincoln, NE, USA). The background was normalized and standardized to the beta-actin housekeeping protein, and the average of three independent experiments was used to carry out statistical analysis.

### 4.10. Microscopic Morphometry of Joint Inflammation

After euthanasia, the legs were fixed in 10% formaldehyde for 2 days, followed by decalcification [60]. Once the joints became soft, they were dehydrated with alcohol and xylol in an ascending concentration. The samples were then embedded in paraffin, sectioned with a microtome (5 µm), and stained with hematoxylin and eosin. After analyzing the stained samples, they were secondly stained for a better representation of inflammation using Safranin-O/Fast Green (Merck KGaA) (Appendix A).

### 4.11. Ethics

The studies were approved by the Ethics Committee of the University of Debrecen, and the humane care of the animals adhered to the “Principles of Laboratory Animal Care” outlined in EU Directive 2010/63/EU (license no. 30/2017/DEMÁB).

### 4.12. Statistical Analysis

Statistical analyses were performed using GraphPad Prism software for Windows, version 9.5.1 (La Jolla, CA, USA). All data are presented as the mean ± standard error of the mean (S.E.M). The Shapiro–Wilk normality test was used to assess the Gaussian distribution. The Kruskal–Wallis test with Dunn’s post-test was employed for non-Gaussian distributed data, while ordinary one- or two-way ANOVA, two-way repeated measures ANOVA, or mixed-effects analysis was used for Gaussian distributed data. Probability values (*p*) less than 0.05 were considered to be significantly different.

## 5. Conclusions

To the best of our knowledge, this is the first comprehensive experimental study to demonstrate that influenza vaccination with a different timing has a low substantial impact on the progression and severity of the inflammatory and related pain responses. Nevertheless, delayed vaccination could alter disease activity, as indicated by the findings from assessments of edema and inflammatory biomarkers. HO-1 might be a severity marker for the inflammatory processes, which is increased by late vaccination. However, revealing the regulation and the role of it needs further mechanistic investigations, which were beyond the scope of the present study. From a translational aspect, it may be worth further investigating this biomarker in relation to flu vaccinations of RA patients to determine the optimal individualized immunization strategies.

## Figures and Tables

**Figure 1 ijms-25-03292-f001:**
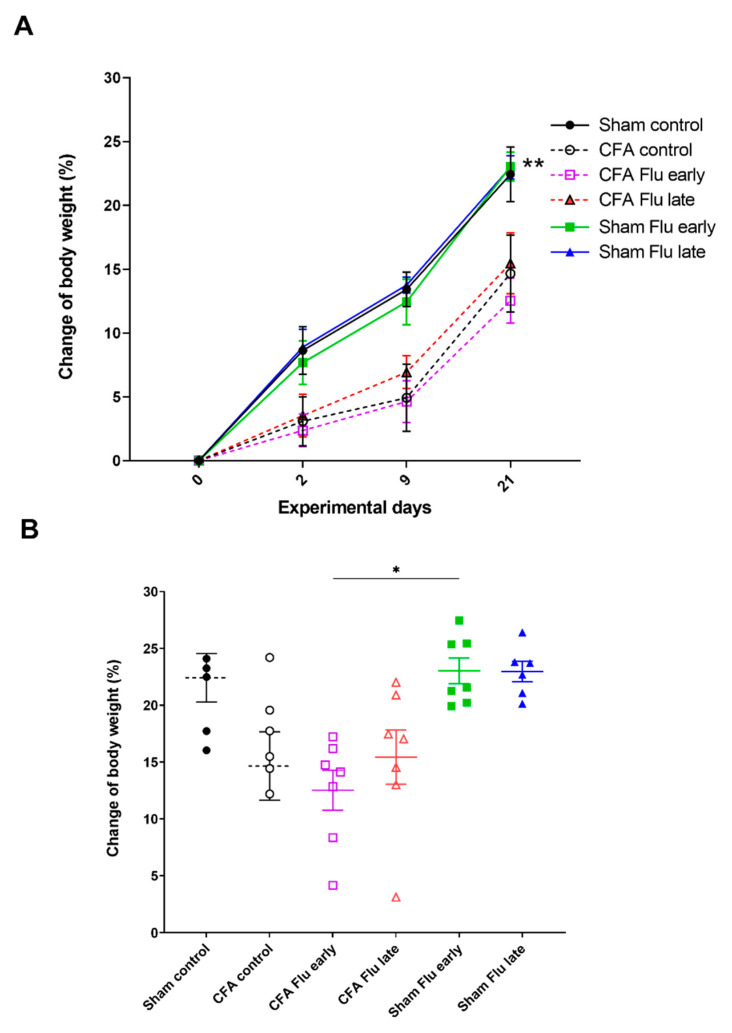
Effect of influenza vaccination on complete Freund’s adjuvant (CFA)-induced body weight loss. (**A**) Measurements were conducted before and 2, 9, and 21 days after CFA injection. Changes in body weight compared to the baseline values were expressed in percentages. Data are shown as means ± S.E.M of *n* = 6–7 rats/group, ** *p* < 0.01 vs. CFA Flu early (two-way repeated measures ANOVA followed by Tukey’s multiple comparison test). (**B**) Comparison of body weight changes on day 21. Data are shown as mean ± S.E.M. of *n* = 6–7 rats/group, * *p* < 0.05 (ordinary one-way ANOVA was performed to determine differences followed by Tukey’s multiple comparison test).

**Figure 2 ijms-25-03292-f002:**
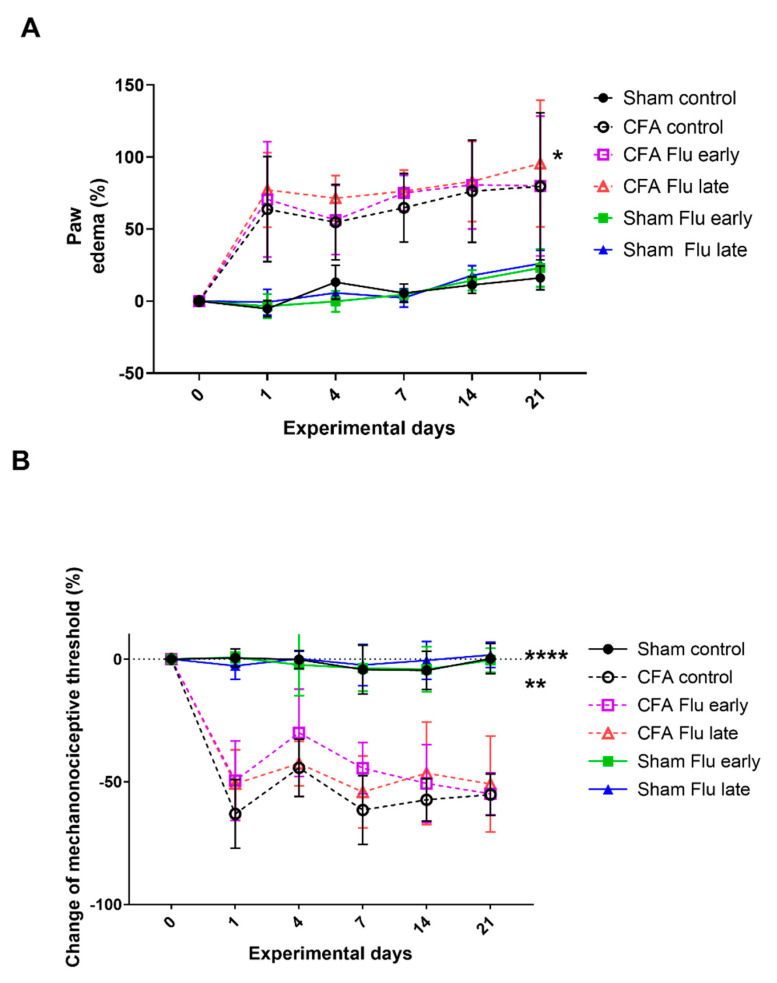
(**A**) Effect of influenza vaccination on complete Freund’s adjuvant (CFA)-induced hind paw edema and (**B**) mechanical allodynia. Data were expressed as a percentage change compared to the baseline measurement’s results. Data are shown as mean ± S.E.M. of *n* = 6–7 rats/group, * *p* < 0.05, ** *p* < 0.01, and **** *p* < 0.0001 vs. respective Sham groups (two-way repeated measures ANOVA followed by Tukey’s multiple comparison test).

**Figure 3 ijms-25-03292-f003:**
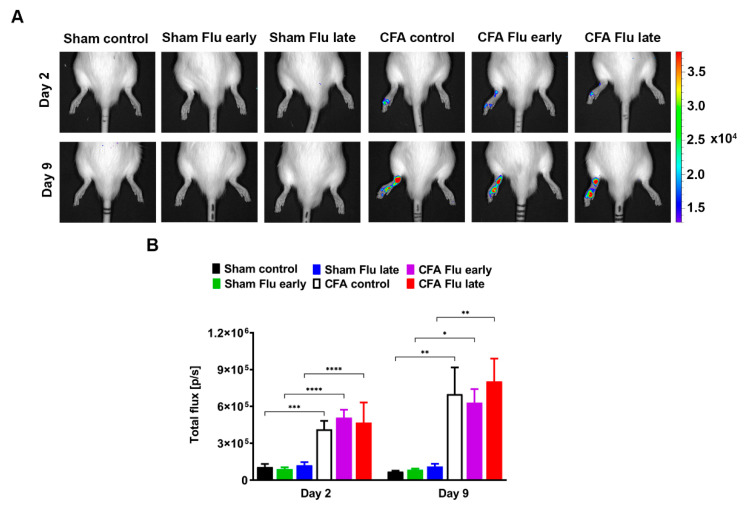
Effect of influenza vaccination on complete Freund’s adjuvant (CFA)-induced neutrophil myeloperoxidase (MPO) activity increase. (**A**) Representative bioluminescence images and (**B**) quantitative analysis of neutrophil MPO activity in the ipsilateral hind paws of non-vaccinated and vaccinated CFA-injected rats as compared to the respective non-arthritic groups 2 and 9 days after arthritis induction. Data are shown as mean ± S.E.M of *n* = 6–7 rats/group, * *p* < 0.05, ** *p* < 0.01, *** *p* < 0.001, and **** *p* < 0.0001 (two-way ANOVA followed by Sidak’s multiple comparison test).

**Figure 4 ijms-25-03292-f004:**
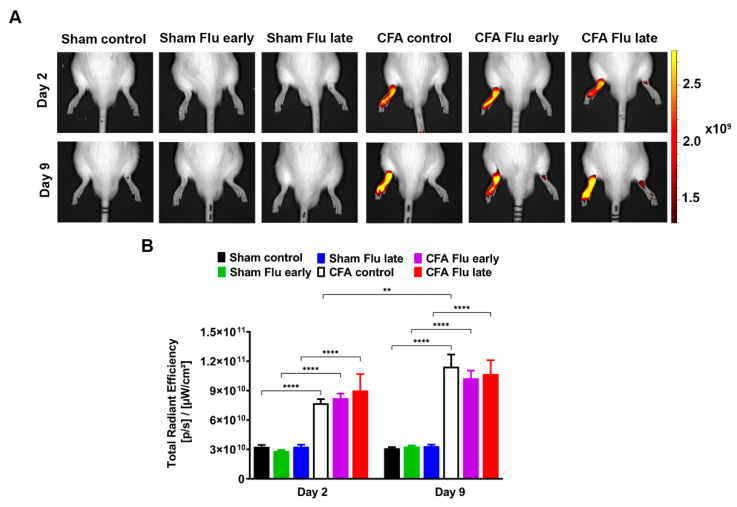
Effect of influenza vaccination on complete Freund’s adjuvant (CFA)-induced plasma extravasation increase. (**A**) Representative fluorescence images and (**B**) quantitative analysis of plasma extravasation in the ipsilateral hind paws of non-vaccinated and vaccinated CFA-injected rats as compared to the respective sham groups 2 and 9 days after arthritis induction. Data are shown as mean ± S.E.M of *n* = 6–7 rats/group, ** *p* < 0.01 (mixed-effects model followed by Sidak’s multiple comparison test), and **** *p* < 0.0001 (two-way ANOVA followed by Sidak’s multiple comparison test).

**Figure 5 ijms-25-03292-f005:**
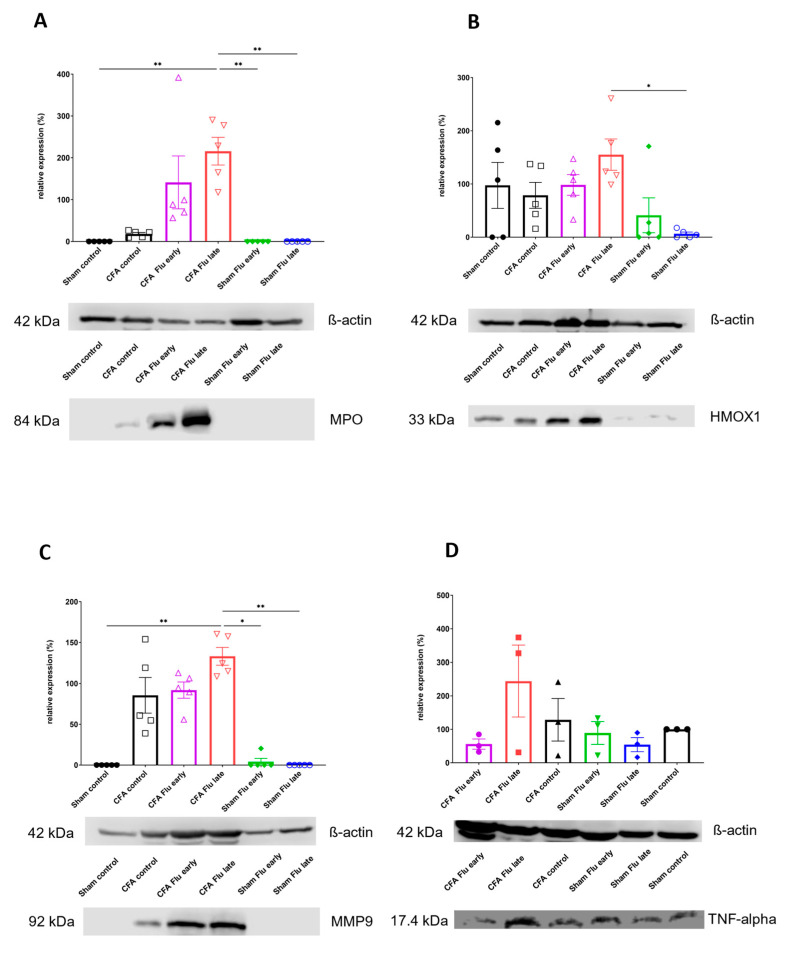
Effect of influenza vaccination on inflammatory biomarkers (MPO, HMOX1, MMP9, TNF-alpha). Results of statistical analysis and a representative image of the Western blot technique of (**A**) MPO, (**B**) HMOX1, (**C**) MMP9, and (**D**) TNF-alpha. Data are shown as mean ± S.E.M of *n* = 6–7 rats/group, * *p* < 0.05, ** *p* < 0.01 (Kruskal–Wallis test followed by Dunn’s multiple comparison test in the case of panel (**A**–**C**) and the ordinary one-way ANOVA followed by Tukey’s multiple comparison test in the case of panel (**D**)).

**Table 1 ijms-25-03292-t001:** Summary of the biomarkers’ diagnostic importance and their correlation with disease activity.

Potential Diagnostic Biomarkers		Diagnostic Biomarkers in Humans According to ACR/EULAR 2010	Correlation with Disease Activity in Humans	Correlation with Disease Activity in Experimental Arthritis	References for Human Biomarkers	References for Experimental Biomarkers
ESR		+	+	+	[15]	[16]
ACPAs		+	−	+	[17,18]	[19,20]
RF		+	−	+	[18,21]	[19,20]
CRP		+	+	+	[22]	[16,20]
MBDA	TNF-α VCAM-1 EGF VEGF-A IL-6 TNF-R1 MMP-1 MMP-3 YKL-40 leptin resistin SAA CRP	−	controversial data	controversial data	[23,24,25,26]	[27,28]
MPO		−	+	+	[29,30]	[31,32]
HMOX-1		−	not known	not known	[33,34]	[35]

ACR/EULAR: American College of Rheumatology/European Alliance of Associations for Rheumatology; ESR: erythrocyte sedimentation rate; ACPAs: Autoantibodies Against Citrullinated Proteins, RF: rheumatoid factor; CRP: C-reactive protein; MBDA: Multi-Biomarker Disease Activity; VCAM-1: vascular cell adhesion molecule-1; EGF: epidermal growth factor; VEGF-A: vascular endothelial growth factor A; IL-6: interleukin 6; TNF-R1: tumor necrosis factor receptor type 1; MMP-1: matrix metalloproteinase-1; MMP-3: matrix metalloproteinase-3l; YKL-40: human cartilage glycoprotein 39; SAA: serum amyloid.

**Table 2 ijms-25-03292-t002:** Summary of the treatment protocol and the experimental groups.

	Sham Control	CFA Control	CFA Early	CFA Late	Sham Flu Early	Sham Flu Late
CFA administration	−	+	+	+	−	−


3Fluart	−	−	+ day 1	+ day 7	+ day 1	+ day 7


+: administration of the representative agent (CFA or 3Fluart vaccine), −: lack of the representative agent’s administration (CFA or 3Fluart vaccine).

## Data Availability

The data that support the findings of this study are available from the corresponding author upon reasonable request. Some data may not be made available because of privacy or ethical restrictions.

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
