# Peer review of "Assessing the Impact of Influenza Vaccination Timing on Experimental Arthritis: Effects on Disease Progression and Inflammatory Biomarkers"

_ijms, 2024, doi:10.3390/ijms25063292_

Round 1

Reviewer 1 Report

Comments and Suggestions for Authors

The authors investigated the effects of influenza vaccination timing on rheumatoid arthritis (RA) progression using a rat model. Findings indicate vaccination during the quiescent phases of RA may exacerbate subclinical inflammation in a time-dependent manner. This well-controlled study suggests myeloperoxidase and heme oxygenase-1 hold promise as indicators of inflammation severity in RA.

I have the following comments:

1. While the study design is generally solid, the sample size of 4 rats per group is quite small. Using more animals would strengthen the statistical power to detect differences between groups.

2. The histology analysis is very limited qualitatively - more descriptive detail of the joint structural changes and immune cell infiltration patterns would be informative.

3. The discussion of heme oxygenase-1 as an inflammation severity marker requires more supporting evidence; this seems speculative based on the data presented. Further studies are needed to substantiate any putative roles of HO-1 as a biomarker prior to making significant claims about clinical utility.

4. In Figures 1–4, the legend is not properly designed and is not consistent with the figures. For example, it is hard to distinguish the groups (baseline control vs. CFA control, CFA flu early vs. flu early). I suggest using a hollow shape for the group in the comparison.

5. In Figure 6, why do some bars have frames in gray while others do not? Please change that. Also, consider switching the β-actin band with the bands of the other markers. It is not necessary to show the exact p-value. Just show that *p<0.05 is enough.

6. There is an abnormal white space in line 335.

Author Response

Answer

TO:

Reviewer 1

Manuscript ID: ijms-2857108

FOR THE ORIGINAL MANUSCRIPT ENTITLED:

Assessing the Impact of Influenza Vaccination Timing on Rheumatoid Arthritis: Effects on Disease Progression and Inflammatory Biomarkers

SUBJECT: Major revision (Tarjanyi, et al.)

We extend our sincere gratitude to the editor and the reviewers for their thorough review and dedication to our manuscript. Their constructive feedback and suggestions have undoubtedly played a significant role in enhancing the quality of our work. We have carefully addressed each of the reviewers’ comments and recommendations in detailed below.

  1. While the study design is generally solid, the sample size of 4 rats per group is quite small. Using more animals would strengthen the statistical power to detect differences between groups.

We agree that this was a potential limitation of the study. We felt that our former bioluminescent and fluorescent imaging required a comprehensive reassessment. For optimisation of detection specifications, and image analysis we involved two professionals (A.I.H.; N.S.) who are experts in this experimental model and along with the in vivo imaging. Due to their extensive professional expertise, they were able to reassess all raw data from the animals included in the experiment, thereby enabling an increase in the number of elements in both bioluminescent and fluorescent imaging investigations. They have made significant contributions to the analysis and interpretation of our data. Please find all of the changes in the main text and supplementary material as well. All modifications in the manuscript have been highlighted in yellow.

  1. The histology analysis is very limited qualitatively - more descriptive detail of the joint structural changes and immune cell infiltration patterns would be informative.

Thank you for expressing your concern about our histology analysis, we highly agree with this limitation of the study. For this reason, the representative images of histology were transferred to the supplementary files since the quality of the microscopic sections are not enough reliable to allow further detailed analysis.

  1. The discussion of heme oxygenase-1 as an inflammation severity marker requires more supporting evidence; this seems speculative based on the data presented. Further studies are needed to substantiate any putative roles of HO-1 as a biomarker prior to making significant claims about clinical utility.

We appreciate the reviewer’s feedback. We have to agree with the reviewer’s assessment, as we drawn overly ambitious conclusion based on our experimental findings. It needs more investigations to confirm the activated heme oxygenase 1 pathway. Consequently, we have revised and greatly moderate the interpretation of our results concerning the heme oxygenase 1 (HO-1) enzyme. We highlighted with yellow the connected sentences in the abstract, discussion and conclusion section. Thank you again for your valuable and correct comment!

  1. In Figures 1–4, the legend is not properly designed and is not consistent with the figures. For example, it is hard to distinguish the groups (baseline control vs. CFA control, CFA flu early vs. flu early). I suggest using a hollow shape for the group in the comparison.

We apologize for the inconsequence of this part of the manuscript, we corrected the figures and their legends according to the expectations of both of the reviewers hoping that it will contribute to better understanding.

  1. In Figure 6, why do some bars have frames in gray while others do not? Please change that. Also, consider switching the β-actin band with the bands of the other markers. It is not necessary to show the exact p-value. Just show that *p<0.05 is enough.

Thank you for identifying this difference between the styles of the frames we apologize and have already made the change.

  1. There is an abnormal white space in line 335.

Thank you for establishing this mistake, we apologize and have corrected it.

Reviewer 2 Report

Comments and Suggestions for Authors

I’m not an expert in animal studies but in my opinion, there is too much confirming that used methodology is indeed suitable for observation of RA developing, especially in Discussion. Given such uncertain and difficult to study topic such as the impact of vaccination on developing autoimmune diseases the methodology should be presented without any doubts.

It seems that Authors have some doubts for the methodology they used if they try to convince readers about it. Also, results are a bit too shortly described to form any conclusions based on them. For example, Authors measured typical markers of RA but also drew conclusions based on body weight or paw volume which could be associated with some other factors.

But the main drawback of this manuscript is the lack of a reference study showing that disease (influenza?) is more harmful for tested RA rats then vaccination. Authors should complement their study with an experiment with inducing influenza in RA rats. Instead, they took as a baseline healthy rats, with neither RA nor influenza. Authors tested only if vaccination can worsen the condition of RA rats but not if vaccination can induce RA or if vaccination is worse or better than viral disease itself and this was the main hypothesis made at the beginning. I doubt if all the above justifies inducing hyperalgesia in animals. There are other possibilities (e.g. biochemical) to test the animal condition without inducing and measuring pain.

Other minor points:

- Discussion should be significantly changed because it includes fragments more fitted to the introduction like, e.g., description of inflammation markers. As a result, it is difficult to spot the main conclusions from the study.

Lines 95-100 – not clear and a bit too cautious

- Colors should be added to plots because lines are hardly distinguishable. 

Comments on the Quality of English Language

Minor editing of English language required.

Author Response

Answer

TO:

Reviewer 2

Manuscript ID: ijms-2857108

FOR THE ORIGINAL MANUSCRIPT ENTITLED:

Assessing the Impact of Influenza Vaccination Timing on Rheumatoid Arthritis: Effects on Disease Progression and Inflammatory Biomarkers

SUBJECT: Major revision (Tarjanyi, et al.)

We genuinely thank the editor and the reviewers for their attention and dedication to our manuscript. Their constructive comments and suggestions have certainly made an important contribution to improving our manuscript. Please, read our point-by-point response to all the reviewers' comments and recommendations.

  1. I’m not an expert in animal studies but in my opinion, there is too much confirming that used methodology is indeed suitable for observation of RA developing, especially in Discussion. Given such uncertain and difficult to study topic such as the impact of vaccination on developing autoimmune diseases the methodology should be presented without any doubts. It seems that Authors have some doubts for the methodology they used if they try to convince readers about it.

Regarding the discussion section of the manuscript, we appreciate your attention to this aspect. While we acknowledge the complexity of studying the impact of vaccination on autoimmune diseases, we remain confident in the methodology employed to induce experimental arthritis in murine models. Based on current knowledge and experimental conditions we believe that the method utilized is well-suited for assessing the effects of vaccines at various stages of inflammation. Additionally, we have sought the expertise of two professionals (A.I.H. and N.S.) in the field to further enhance the rigor and interpretation of our data.

To ensure transparency and clarity, all revisions to the manuscript have been highlighted in yellow, and we have utilized the “Track Changes” function for easy identification of modifications throughout the text. Thank you once again for your valuable feedback, which has been instrumental in refining our work.

Snekhalatha, U., et al., Evaluation of complete Freund's adjuvant-induced arthritis in a Wistar rat model. Comparison of thermography and histopathology. Z Rheumatol, 2013. 72(4): p. 375-82.

van den Berg, W.B., et al., Amelioration of established murine collagen-induced arthritis with anti-IL-1 treatment. Clin Exp Immunol, 1994. 95(2): p. 237-43.

Roy, Tanushree, and Saikat Ghosh. "Animal models of rheumatoid arthritis: correlation and usefulness with human rheumatoid arthritis." Indo American Journal of Pharmaceutical Research 3.8 (2013): 6131-6142.

  1. Also, results are a bit too shortly described to form any conclusions based on them. For example, Authors measured typical markers of RA but also drew conclusions based on body weight or paw volume which could be associated with some other factors.

Thank you for pointing this out. We complemented the results section according to the requirement of the reviewer. We completely agree with the reviewer’s opinion about the fact that many factors can influence body weight or cause an increase in paw volume. For this reason, we examined other more specific parameters too, to prove our hypothesis (in vivo measurements, western blot analysis of specific proteins), also we would like to highlight that our qualified co-workers in the Animal House monitored the animals daily to be able to exclude rats with different disorders from the study fitting the standards of the Ethics Committee.

In response to this observation, the following statement has been included in the discussion section: “While body weight measurement alone may not serve as a definitive marker for CFA-induced arthritis or hold independent diagnostic value, they represent one of many parameters in our observations, aligning with findings from previous studies.”

  1. But the main drawback of this manuscript is the lack of a reference study showing that disease (influenza?) is more harmful for tested RA rats then vaccination. Authors should complement their study with an experiment with inducing influenza in RA rats. Instead, they took as a baseline healthy rats, with neither RA nor influenza. Authors tested only if vaccination can worsen the condition of RA rats but not if vaccination can induce RA or if vaccination is worse or better than viral disease itself and this was the main hypothesis made at the beginning.

We highly agree with this point of view since it is crucial in the vaccine approval process to prove, that the vaccine is less harmful than the infection itself. However, according to our latest knowledge, there is, unfortunately, no opportunity to study the pathomechanisms of influenza infection and rheumatoid arthritis in the same animal strain. The rat strain used for this present study is prone to be infected with influenza, but the pathological changes which occur in humans are not observable. Confirming this before the beginning of the study we carefully investigated the latest research articles on this topic, and find out, that maybe cotton rats would fit our expectations, but the price of this strain of the laboratory animal exceeded our funding frame.

The other reason we chose not to examine the effects of influenza was based on literature data, which confirms that influenza vaccination reduces morbidity and mortality of rheumatoid arthritis patients, we aimed to establish the ideal time point of vaccination instead of deciding whether to vaccinate or not. To strengthen this viewpoint, we inserted further citations into the Introduction part. Again, thank you for your suggestion, it would have been interesting to explore this aspect.

Bouvier NM, Lowen AC. Animal Models for Influenza Virus Pathogenesis and Transmission. Viruses. 2010;2(8):1530-1563. doi: 10.3390/v20801530. PMID: 21442033; PMCID: PMC3063653.

Choudhary N, Bhatt LK, Prabhavalkar KS. Experimental animal models for rheumatoid arthritis. Immunopharmacol Immunotoxicol. 2018 Jun;40(3):193-200. doi: 10.1080/08923973.2018.1434793. Epub 2018 Feb 12. PMID: 29433367.

Chen CM, Chen HJ, Chen WS, Lin CC, Hsu CC, Hsu YH. Clinical effectiveness of influenza vaccination in patients with rheumatoid arthritis. Int J Rheum Dis. 2018 Jun;21(6):1246-1253. doi: 10.1111/1756-185X.13322. PMID: 29879317.

Furer, V., et al., 2019 update of EULAR recommendations for vaccination in adult patients with autoimmune inflammatory rheumatic diseases. Ann Rheum Dis, 2020. 79(1): p. 39-52.

Dirven, L., T.W. Huizinga, and C.F. Allaart, Risk factors for reported influenza and influenza-like symptoms in patients with rheumatoid arthritis. Scand J Rheumatol, 2012. 41(5): p. 359-65.

  1. I doubt if all of the above justifies inducing hyperalgesia in animals. There are other possibilities (e.g. biochemical) to test the animal condition without inducing and measuring pain.

We appreciate your acknowledgement of our commitment to the well-being of the animals involved in the study. Our efforst to identify a new specific biomarker for monitoring disease progression underscore our dedication to this principle. Unfortunately, it was necessary to observe painful symptoms of the disease as part of our research, given that the treatment and assessment of rheumatoid arthritis still heavily rely on subjective patient experiences (e.g., Disease Activity Score, Health Assessment Quastionare Disability Index, Rheumatoid Arthritis Quality of Life Questionare).

We want to assure the reviewer that our research team conducted all experiments with the utmost respect for animal welfare, adhering to the guidelines set forth by the Ethical Committee and following the principles of the 3R rule (reduction, replacement, refinement). As a demonstration of our commitment to animal well-being, we have included a new line in the manuscript:

“When the rat removed its paw, the machine immediately stopped, preventing further discomfort or tissue damage, and the allodynia numerical value was recorded from the digital display.”

Furthermore, our study utilized the Dynamic Plantar Aesthesiometer, a sophisticated system that automates the application of force through an electromagnetic, silent motor ranging from 0 to 100 grams, adjustable at a rate of 0 to 50 seconds. In our experimental protocol, animals were subjected to 50 grams of force for 10 seconds. It is important to note that this force level typically represent a non-noxious stimulus, hence the use of the term “allodynia” throughout the main text. The term “hyperalgesia” has been appropriately corrected to “allodynia” in all instances throughout the manuscript.

  1. Other minor points:
  2. Discussion should be significantly changed because it includes fragments more fitted to the introduction like, e.g., description of inflammation markers. As a result, it is difficult to spot the main conclusions from the study.

Thank you for your analysis, it is essential for the reader's better understanding. To further enhance the quality of our manuscript, we have also involved two professionals whose contribution hopefully changed the perspective of the discussion significantly. As a result, some points of the Discussion were moved into the Introduction part of the manuscript, and some sentences were clarified as well.

Lines 95-100 – not clear and a bit too cautious

As a consequence of the above-mentioned adjustment in the author's list, the lines (formerly) were deleted and revised also.

Round 2

Reviewer 1 Report

Comments and Suggestions for Authors

The authors have made their best efforts to address my comments from the previous round of review. I have no more suggestions. 

Author Response

Answer

TO:

Reviewer 1

Manuscript ID: ijms-2857108

FOR THE ORIGINAL MANUSCRIPT ENTITLED:

Assessing the Impact of Influenza Vaccination Timing on Experimental Arthritis: Effects on Disease Progression and Inflammatory Biomarkers

SUBJECT: Minor revision (Tarjanyi, et al.)

We are writing to express our sincere gratitude for taking the time to review our paper. Your insightful feedback and constructive criticism have been invaluable in shaping and improving the quality of our work. Your expertise and meticulous attention to detail have greatly contributed to the enhancement of the content.

Warm regards,

The Authors:

Vera Tarjányi, Ákos Ménes, Leila Hamid, Andrea Kurucz, Dániel Priksz, Balázs Varga, Rudolf Gesztelyi, Rita Kiss, Ádám István Horváth, Nikolett Szentes, Zsuzsanna Helyes, Zoltán Szilvássy, Mariann Bombicz

Reviewer 2 Report

Comments and Suggestions for Authors

Authors have answered all my comments, though no additional experiments have been performed. Nevertheless, the financial reason fully justifies it. What still could be more clarified is the main conclusion. Late vaccination (or vaccination during the active disease) seems to affect the severity of the disease, but Authors suggest that rather not. It seems that all is due to the biomarker that is taken to assess the progress of RA. Maybe it would be good to clarify it in Introduction, e.g., by providing a table with all known (and newly suggested like HO-1) biomarkers. It would certainly bring important insights to such difficult disease to assess in terms of patient condition as RA.

Author Response

Answer

TO:

Reviewer 2

Manuscript ID: ijms-2857108

FOR THE ORIGINAL MANUSCRIPT ENTITLED:

Assessing the Impact of Influenza Vaccination Timing on Experimental Arthritis: Effects on Disease Progression and Inflammatory Biomarkers

SUBJECT: Minor revision (Tarjanyi, et al.)

We extend our sincere gratitude to the reviewers for their thorough review and dedication to our manuscript. Their constructive feedback and suggestions have undoubtedly played a significant role in enhancing the quality of our work. We have carefully addressed the reviewer’s comments and recommendations in detail below.

  1. The authors have answered all my comments, though no additional experiments have been performed. Nevertheless, the financial reason fully justifies it.

Thank you for your comprehension.

  1. What still could be more clarified is the main conclusion. Late vaccination (or vaccination during the active disease) seems to affect the severity of the disease, but Authors suggest that it rather not. It seems that all is due to the biomarker that is taken to assess the progress of RA.

Thank you for pointing this out. The reviewer is correct, and we have made the changes in the main conclusion as follows:

„To the best of our knowledge, this is the first comprehensive experimental study to demonstrate that influenza vaccination with different timing has a low substantial impact on the progression and severity of the inflammatory and related pain responses. Nevertheless, delayed vaccination could alter the disease activity, as indicated by the findings from assessments of edema and inflammatory biomarkers. HO-1 might be a severity marker for the inflammatory processes, which is increased by the late vaccination.”

  1. Maybe it would be good to clarify it in Introduction, e.g., by providing a table with all known (and newly suggested like HO-1) biomarkers. It would certainly bring important insights to such difficult disease to assess in terms of patient condition as RA.

Thank you for this excellent suggestion. Accordingly, we included a new table (Table 1.) in the Introduction section to evaluate potential biomarkers. In alignment with recent literature, we correlated them with disease activity in both human subjects and experimental arthritis. Moreover, we incorporated into the same section a brief summary of the potential role of HO-1 between diagnostic biomarkers for identifying RA.

Thank you once again for your expertise and guidance have been instrumental in the development of this work.

Best Regards,

The Authors:

Vera Tarjányi, Ákos Ménes, Leila Hamid, Andrea Kurucz, Dániel Priksz, Balázs Varga, Rudolf Gesztelyi, Rita Kiss, Ádám István Horváth, Nikolett Szentes, Zsuzsanna Helyes, Zoltán Szilvássy, Mariann Bombicz
